# Leptin Reduces Plin5 m^6^A Methylation through FTO to Regulate Lipolysis in Piglets

**DOI:** 10.3390/ijms221910610

**Published:** 2021-09-30

**Authors:** Dongqin Wei, Qian Sun, Yizhou Li, Chaowei Li, Xinjian Li, Chao Sun

**Affiliations:** 1College of Animal Science and Technology, Northwest A&F University, Xianyang 712100, China; weidongqin@nwafu.edu.cn (D.W.); sunqian1020@nwafu.edu.cn (Q.S.); liyizhou@nwafu.edu.cn (Y.L.); cwl123@nwafu.edu.cn (C.L.); 2College of Animal Science and Veterinary Medicine, Henan Agricultural University, Zhengzhou 450002, China

**Keywords:** leptin, Perilipin5, m^6^A methylation, lipolysis, energy metabolism

## Abstract

Perilipin5 (Plin5) is a scaffold protein that plays an important role in lipid droplets (LD) formation, but the regulatory effect of leptin on it is unclear. Our study aimed to explore the underlying mechanisms by which leptin reduces the N^6^-methyladenosine (m^6^A) methylation of Plin5 through fat mass and obesity associated genes (FTO) and regulates the lipolysis. To this end, 24 Landrace male piglets (7.73 ± 0.38 kg) were randomly sorted into two groups, either a control group (Control, *n* = 12) or a 1 mg/kg leptin recombinant protein treatment group (Leptin, *n* = 12). After 4 weeks of treatment, the results showed that leptin treatment group had lower body weight, body fat percentage and blood lipid levels, but the levels of Plin5 mRNA and protein increased significantly in adipose tissue (*p* < 0.05). Leptin promotes the up-regulation of FTO expression level in vitro, which in turn leads to the decrease of Plin5 M^6^A methylation (*p* < 0.05). In in vitro porcine adipocytes, overexpression of FTO aggravated the decrease of M^6^A methylation and increased the expression of Plin5 protein, while the interference fragment of FTO reversed the decrease of m^6^A methylation (*p* < 0.05). Finally, the overexpression in vitro of Plin5 significantly reduces the size of LD, promotes the metabolism of triglycerides and the operation of the mitochondrial respiratory chain, and increases thermogenesis. This study clarified that leptin can regulate Plin5 M^6^A methylation by promoting FTO to affect the lipid metabolism and energy consumption, providing a theoretical basis for treating diseases related to obesity.

## 1. Introduction

Research on how leptin relieves obesity has made great progress since its discovery in 1994. Leptin is a kind of adipokines encoded by the ob gene of adipose tissue, which can effectively regulate and help maintain the body’s physiological energy balance and body weight [1,2]. Simultaneously, there have been some reports about leptin regulating glucose and lipid metabolism, immunity, myogenesis, bone physiology and reproduction [3,4,5,6,7]. Leptin therapy has been approved for hypothalamic amenorrhea, partial lipodystrophy, diabetes, neurodegenerative diseases, depression, and common obesity where plasma leptin levels are relatively low [8]. However, some obese individuals do not have a shortage of leptin, on the contrary their obesity is often related to the phenomenon of “leptin resistance” [9]. Therefore, in-depth study of the mechanism of leptin will contribute to potential future applications of leptin in clinical practice. Furthermore, increasing fat percentage and distribution through leptin is also an effective strategy to improve pork quality.

In recent years, studies have reported that leptin leads to the up-regulation of FTO [10], which inhibits m^6^A methylation modification in the body [11]. FTO was previously recognized to be associated with the occurrence and development of childhood and adult obesity and type 2 diabetes (T2D) [12,13]. Later, some researchers reported that FTO belonged to demethylase and participates in the addition of M^6^A modification sites [12]. There are more than 100 RNA modifications reported in eukaryotes. Among them, M^6^A methylation has an important role in stabilizing mRNA and has attracted more and more attention. For example, zinc finger protein (Zfp 217) depends on m^6^A modification to coordinate transcription and post-transcriptional regulation to promote adipogenic differentiation in mice [14]. However, studies on pigs have shown that FTO inhibits adipogenesis through the molecular mechanism of m^6^A methylation in the post-transcriptional regulation of the JAK2-STAT3-C/EBPβ signal axis [15]. Therefore, M^6^A methylation affects fatty acid biosynthesis metabolism through different targets, and the interaction mechanism between leptin and M^6^A methylation is also worthy of further elucidation.

The Perilipin family is composed of five members (PLIN 1–5), which play a major role in controlling the triacylglycerides hydrolysis and lipolysis in adipose tissue [16]. Plin5 is a scaffold protein in this family that affects the formation of LD. With the continuous deepening of research, LD are currently considered a key manipulator for storing neutral lipids, regulating metabolism, membrane transport, protein degradation and signal transduction in the interaction with other organelles [17]. When the lipolysis of LD in the body is too strong, causing excessive accumulation of FFA in the cytoplasm, Plin5 can reduce the endoplasmic reticulum stress and cell inflammation caused by this phenomenon in mice [18,19,20]. Plin5^-/-^ mice showed a relative decrease in triglyceride content, an increase in fatty acid β-oxidation, and an increase in ROS levels [21]. Another role that cannot be ignored is that Plin5 can act as a key protein in the interaction between mitochondria and LD, affecting mitochondrial biological functions and oxidative stress [22,23]. Plin5 is the core of lipid homeostasis in skeletal muscle, liver, and heart [24]. Therefore, it can be regarded as a key factor affecting the occurrence and development of diseases such as non-alcoholic fatty liver, insulin resistance, and myocardial hypertrophy. However, there are no relevant reports about whether leptin and FTO can effectively regulate the expression of Plin5. Further studies on the m^6^A methylation modification of Plin5 have not been reported. Here, we hypothesize that leptin can regulate the M^6^A methylation of Plin5 through FTO.

In order to confirm this hypothesis, leptin recombinant protein was used to treat piglet or pig primary adipocytes, and the m^6^A methylation modification of Plin5 was used as a point of penetration to explore the interaction mechanism of leptin and Plin5 to regulate lipid metabolism. This provides a nutritional strategy for reducing pig backfat and improving meat quality, and it also brings hope for the treatment and prevention of human related diseases with obesity by regulating the expression of Plin5.

## 2. Results

### 2.1. Leptin Up-Regulates the Expression of Plin5 to Promote Pig Lipolysis

Leptin treatment for 4 weeks significantly increased the levels of leptin protein and leptin receptor mRNA in subcutaneous adipose tissue from piglets (*p* < 0.05, Figure 1A,B). Correspondingly, compared with the control group, the weight, back fat and intramuscular fat content of piglets in the leptin group were significantly lower (*p* < 0.05, Figure 1C–E). The serum triglyceride, cholesterol and low-density lipoprotein contents of piglets in the leptin group were significantly lower than those of the control group, while the high-density lipoprotein contents and lipase were significantly higher than those of the control group (*p* < 0.05, Figure 1F). The body temperature of piglets after leptin treatment was significantly higher than that of the control group (*p* < 0.05, Figure 1G). Furthermore, the mRNA and protein expression of Plin5 in the subcutaneous adipose tissue of the leptin group were significantly increased (*p* < 0.05, Figure 1H,I). The results of immunohistochemistry and Western Blot were consistent (Figure 1J).

In order to further analyze the effects of leptin treatment on pig lipolysis and mitochondrial function, we tested the mRNA and protein levels of lipolysis genes. Leptin treatment significantly increased the mRNA and protein levels of peroxisome proliferator-activated receptor gamma (PPARγ), adipose triglyceride lipase (ATGL), hormone-sensitive lipase (HSL) and lipoprotein lipase (LPL) (*p* < 0.05, Figure 1K and Figure 2A). However, qPCR analysis of fatty acid synthesis-related genes acetyl-CoA carboxylase 1 (*ACC1*) and fatty acid synthase (*Fasn*) found that the leptin group was significantly down-regulated (*p* < 0.05, Figure 2A). Simultaneous, leptin treatment significantly increased the mRNA expression levels of mitochondrial complex-related genes (*p* < 0.05, Figure 2B). The oil red O staining results showed that the number of LD increased, while the volume of LD decreased (Figure 2C). The above results indicate that leptin treatment of piglets can up-regulate Plin5 mRNA and protein levels, enhance lipolysis, and improve mitochondrial function.

### 2.2. Leptin Up-Regulates FTO in Adipocytes to Inhibit m^6^A Methylation of Plin5

In order to verify the above results, we cultured primary porcine preadipocytes and induced their differentiation in vitro and treated the cells with 50 nmol/μL leptin for 24 h to detect the expression of related genes. The results showed that leptin treatment significantly increased the expression of Plin5 mRNA and protein in adipocytes, and further increased the expression of lipolysis-related genes (*p* < 0.05, Figure 2D,E). Immunofluorescence observation showed that the expression of Plin5 was more in the leptin treatment group, and Plin5 was localized around the mitochondria (Figure 2G). It was revealed by staining with BODIPY that there were fewer LD in leptin-treated adipocytes (Figure 2H). The above results indicate that leptin has the same effect in porcine adipocytes and piglet tissues.

To further examine the effect of leptin on M^6^A methylation, we evaluated RNA m^6^A methylation-related genes in pig subcutaneous adipose tissue. The results showed that leptin treatment up-regulated *FTO* and *Ythdf2* (M^6^A methylation-recognized protein) expression (*p* < 0.05, Figure 3A). Furthermore, western blot analysis found that leptin treatment also significantly increased the level of FTO protein (*p* < 0.05, Figure 3B). Dot Blot and M^6^A methylation detection kit were used to determine the level of m^6^A methylation, which showed that the total M^6^A methylation of leptin-treated cells was significantly down-regulated (*p* < 0.05, Figure 3C,D). Finally, the m^6^A level of Plin5 detected by m^6^A-IP was also significantly down-regulated (*p* < 0.05, Figure 3E). Our data preliminarily indicate that leptin reduces the M^6^A methylation of total RNA and Plin5 by promoting the expression of FTO.

Next, we explored whether leptin inhibits M^6^A methylation of Plin5 through FTO. First, we used pc-FTO and si-FTO to transfect porcine adipocytes in vitro and tested the efficiency of the vector. The results showed that the FTO overexpression vector and interference fragments can meet the requirements of subsequent experiments (Figure 3F). Overexpression of FTO in porcine adipocytes significantly reduced the m^6^A methylation level of Plin5 (*p* < 0.05), while interference fragments reversed this effect (Figure 3G). Then, we used leptin and the FTO vector to treat pig adipocytes together, and found that overexpression of FTO aggravated the decrease in M^6^A methylation level of total mRNA (*p* < 0.05), and interference with FTO reduced the corresponding m^6^A methylation (*p* < 0.05, Figure 3H). The m^6^A methylation of Plin5 mRNA was consistent with the change of the total m^6^A methylation level (*p* < 0.05, Figure 3I). These results indicate that leptin treatment inhibits the m^6^A methylation level of total mRNA and Plin5 mRNA, and this effect is caused by the up-regulation of FTO.

### 2.3. FTO Inhibits m^6^A Methylation of Plin5 to Up-Regulate Plin5 Protein Levels

Since leptin regulates Plin5 M^6^A methylation through FTO, we speculate that FTO directly affects Plin5 M^6^A methylation modification to regulate its protein expression. To prove this hypothesis, we transfected porcine adipocytes with vectors that overexpress and interfere with FTO. The results showed that, unlike in leptin treatment, the expression of Plin5 protein in the pc-FTO group was significantly higher (*p* < 0.05, Figure 4B), but there was no change in mRNA level (*p* > 0.05, Figure 4A). This shows that the increase of Plin5 protein expression after leptin treatment is not only the regulation of transcription level, but also the post-transcriptional modification, that is, the regulation of m^6^A methylation modification.

To verify that the expression of Plin5 is indeed regulated by m^6^A methylation modification, we used the m^6^A methylation inhibitor Cycloleucine (CL) and the agonist Betaine (Bet) to treat porcine adipocytes. First of all, we found that CL treatment significantly reduced cell M^6^A methylation levels (*p* < 0.05), and Bet extremely significantly increased M^6^A methylation levels (*p* < 0.01, Figure 4C). When overexpression of or interference with FTO, the treatment of CL and Bet significantly affected the total M^6^A methylation level and Plin5 protein expression (*p* < 0.05, Figure 4D,F,H), and also had no effect on the mRNA level (*p* > 0.05, Figure 4E,G). This result suggests that M^6^A methylation of Plin5 is a key factor in regulating its protein expression.

In order to further determine whether m^6^A methylation regulates the expression of Plin5, we analyzed the m^6^A modification sites of Plin5 mRNA on the SRAMP website [25]; a total of 23 sites (Figure 4I). We selected the site closest to the stop codon at the 3′ UTR end for mutation, and constructed a Plin5 methylation site mutation vector (Figure 4J). The luciferase activity test showed that the 3′UTR end of Plin5 was successfully mutated (Figure 5A). Further, we used this vector to transfect cultured adipocytes in vitro and showed that the total M^6^A methylation level of the Mut grouTablep overexpressing FTO did not change significantly (*p* > 0.05, Figure 5B), and the mRNA level of Plin5 did not change significantly (*p* > 0.05, Figure 5C). Compared with the WT, the methylation level of Plin5 M^6^A after the mutation is down-regulated to a lower degree (*p* < 0.05, Figure 5D). The expression of Plin5 protein is also consistent with the changes of its M^6^A methylation level (*p* < 0.05, Figure 5E). The above results support our hypothesis.

### 2.4. Plin5 Promotes the Breakdown of LD to Improve Lipolysis and Energy Metabolism

In order to further evaluate the role of Plin5 in lipid metabolism, we constructed a Plin5 overexpression vector. Treatment of cultured preadipocytes with this vector showed that the level of Plin5 mRNA increased significantly (*p* < 0.001, Figure 5F). Next, we induced the differentiation of primary porcine preadipocytes under Plin5 overexpression. The differentiated cells were ultracentrifuged to extract LD (Figure 5H), and their size was marked by BODIPY fluorescent staining. The results showed that the size of LD in the pc-Plin5 group was significantly lower than that in the control group (*p* < 0.05, Figure 5G). The overexpression of Plin5 further increased the mRNA and protein expression of lipid droplet protein genes *ATGL*, *HSL*, *ADBH5* and lipolysis-related genes *LPL* and carnitine palmitoyltransferase 1A (*CPT1a*) (*p* < 0.05 Figure 5I,J). Consistently, the mRNA expression levels of the thermogenesis genes *PGC1α*, *UCP3*, *Atp5a1* and mitochondrial function genes *TFAM*, *Ndufb8*, *Sdhb*, *Uqcrc2*, *Cox4i1* in the pc-Plin5 group were also significantly increased (*p* < 0.05, Figure 6A,B). Furthermore, we analyzed the protein levels of genes that were extremely increased in the pc-Plin5 group and found that overexpression of Plin5 also significantly increased the protein levels of PPARGC1A and NDUFB8 (*p* < 0.05, Figure 6C). We found that Plin5 also has an effect on energy metabolism, because the ATP production and NAD^+^/NADH ratio of the overexpression Plin5 group were also significantly higher than in the control group (*p* < 0.05, Figure 6D,E). These results indicate that the up-regulation of Plin5 can promote the action of lipolytic enzymes on the surface of LD to accelerate the degradation of LD, promote the operation of the mitochondrial respiratory chain, and increase heat production and energy metabolism.

### 2.5. Plin5 Promotes Lipid Synthesis and Enhances Mitochondrial β-Oxidation under Lipotoxicity Models

In order to study the role of Plin5 in fatty acid disorders, we used palmitic acid (PAL) to construct a fatty acid lipotoxicity model. PAL treatment in vitro had no effect on the mRNA levels of apoptosis genes such as *Casp3*, *Casp9*, *Bax*, and *Bcl-2* (*p* > 0.05, Figure 6F). Similarly, Tunel staining showed that there was no significant increase in the number of apoptotic cells after PAL treatment (*p* > 0.05, Figure 6G). It indicated that PAL treatment did not cause apoptosis of porcine adipocytes but caused a very significant increase in cellular FFA levels (*p* < 0.01, Figure 6H). BODIPY fluorescent staining shows accumulation of LD (Figure 6I). These data indicate that our lipotoxicity model was successfully constructed. Furthermore, we used the Plin5 overexpression vector to transfect cells under this model and found that FFA and NAD^+^/NADH were significantly reduced (*p* < 0.05, Figure 6J, *n*). Interestingly, the overexpression of Plin5 under PAL treatment further increased the expression of β-oxidation rate-limiting enzyme *ACSL2* and triglyceride synthase *DGAT2*, *Fasn*, and *ACC1* (*p* < 0.05, Figure 6K). Mitochondrial complex-related genes are also significantly up-regulated (*p* < 0.05, Figure 6L). Further, the relative protein expression level was consistent with gene expression (Figure 6M). Our results indicate that Plin5 can alleviate the metabolic disorder caused by excessive FFA in the body. Under the lipotoxicity model, Plin5 can promote lipid anabolism and enhance mitochondrial β-oxidation.

## 3. Discussion

Leptin can effectively regulate body fat deposition and body weight through a variety of mechanisms [3,26]. In this study, the treatment of piglets with leptin recombinant protein showed that fat cells became smaller, the size and number of LD became smaller, and the expression of lipolytic genes increased, which is consistent with the results of other studies. In porcine adipocytes, leptin mainly activates leptin Receptor B to act on downstream JAK2 signaling molecules. JAK2 binds to the specific binding region of the C-terminal of leptin Receptor B, leading to the phosphorylation and activation of JAK2/STAT3 [27]. Phosphorylated STAT3 enters the nucleus to regulate the expression of transcription factors such as the PPAR family and the PI3K and AMPK signaling pathways [28]. Studies have shown that PI3K/PPARα is a key signal regulating the Plin5 family [19]. Combined with the results of previous studies, we found that leptin can regulate the expression of Plin5 mRNA and protein, but its mechanism of action still needs further study. In addition, leptin also has other effects on regulating lipolysis. For example, leptin induces lipolysis of white adipose tissue through nitric oxide, and the leptin-adiponectin axis is involved in regulating dysfunctional adiposity [29,30]. Our results indicate that leptin-mediated Plin5 has a significant effect on promoting lipolysis and enhancing mitochondrial function.

Our laboratory previously reported that leptin regulates the JAK2/STAT3 signaling pathway through SOCS3, leading to the up-regulation of Plin5 expression [31]. It is shown in our data that leptin treatment has increased the mRNA and protein expression of PPARγ, HSL, ATGL, and LPL. Other reports have found that PPARγ induces the accumulation of Plin5 [32], and further HSL and ATGL can interact with Plin5 to regulate lipolysis [22,33]. Interestingly, we found through preliminary experiments that the JAK2/STAT3 signaling pathway inhibitor (SD1008) and leptin co-treatment of 3T3-L1 did not cause changes in the expression of Plin5 mRNA, but the protein level of Plin5 was still significantly up-regulated. This indicates that the up-regulation of Plin5 after leptin treatment is not only regulated by the JAK2/STAT3 pathway and lipolytic gene, and implies that there are other ways of regulation. In addition, leptin can up-regulate the expression of FTO [34], which is one of the enzymes that catalyzes the demethylation of m^6^A in vivo [35], indicating that leptin may also regulate the expression of Plin5 by affecting m^6^A methylation. This study found that leptin up-regulated the expression of FTO and YTHDF2 in pig subcutaneous adipose tissue and decreased the level of RNA m^6^A methylation. The results of FTO overexpression and interference fragment treatment are also consistent with our hypothesis. This confirms that leptin can inhibit the m^6^A methylation of Plin5 by up-regulating the expression of FTO. However, the relationship between the decrease of m^6^A methylation of Plin5 and the regulation of Plin5 expression needs further experiments to prove.

Our results show that the decrease of m^6^A methylation level has no effect on Plin5 transcription, but it will lead to enhanced translation and increase the expression of Plin5 protein. In order to exclude the possible influence of FTO itself on the expression of Plin5, we used CL and Bet to treat the cells. CL is a small molecule inhibitor of methylation, which can effectively reduce the level of m^6^A methylation [36]; while Bet is the methyl donor of m^6^A methylation in the body [37], which can significantly increase the level of methylation in the body. The results showed that it was consistent with the FTO vector treatment. CL significantly reduced the m^6^A methylation level of Plin5 and increased the protein expression of Pelin5; the Bet treatment was the opposite of CL. After further mutation of the Plin5 methylation site, overexpression of FTO did not change the m^6^A methylation level of Plin5, and its protein level did not increase. Previous studies have found that decreased m^6^A methylation will increase protein levels [38,39]. Our experiments confirmed that the up-regulation of Plin5 protein expression is caused by FTO demethylation. However, there are reports that obesity-associated SNPs are not functionally related to *FTO*, but are related to *FTO* neighboring genes *IRX 3* and *RPGRIP1L* [40]. Although the interference of FTO affects body weight and composition [10,11], FTO intron variants and their activity are still elusive. *IRX 3* and *RPGRIP1L* have not been reported to regulate methylation. Our results prove that M^6^A methylation of Plin5 is related to FTO, and whether there are other intron variants is a new direction worthy of our further exploration.

Under normal physiological conditions, the body is in a fine-tuned and relatively stable environment, which we call homeostasis. When the body receives external stimuli, the homeostasis will be destroyed, which will lead to metabolic disorders and various diseases. For example, in obesity, the increase in FFA content triggers lipid metabolism disorders and chronic low-grade inflammation, thereby causing diseases such as insulin resistance and non-alcoholic fatty liver [41]. Given the strong association between the excess of weight and T2D, the focus of a suitable antidiabetic treatment of obese patients should at least be the prevention of additional weight gain [42]. Previous studies have found that there are a variety of proteins on the surface of LD, which are involved in a variety of regulatory processes, such as lipid metabolism, energy metabolism, membrane transport, protein degradation, immune response, and transcriptional regulation [17]. The Plin family is likely to be the key factor that makes these LD proteins function [16]. It is reported that Plin5 and ABHD5, HSL, ATGL co-regulate the breakdown of LD and lipid metabolism [33,43]. Our results show that the overexpression of Plin5 can promote LD degradation and activate downstream lipolytic genes and mitochondrial functional gene expression; while in a lipotoxic state, Plin5 can promote lipid anabolism and fatty acid β-oxidation, and accelerate FFA consumption. Under lipid stimulation, Najt et al. [18] characterized Plin5 as a fatty acid binding protein that preferentially binds to LD-derived monounsaturated fatty acids (MUFA) and transfers to the nucleus under cAMP/PKA mediation to promote lipolysis. Other studies have revealed that Plin5 can reduce FFA-induced metabolic disorders and inflammation, mainly because Plin5 relies on sirtuin 1 to promote autophagy [44]. On the other hand, the contact of Plin5 with mitochondrial-LD to improve mitochondrial function is also one of the reasons for its potential role [23]. By revealing the function of Plin5, we provide a target for the treatment of diseases related to obesity, such as T2D, in the future and also provide a basis for studying the interaction between LD and other organelles.

## 4. Material and Methods

### 4.1. Animals and Samples

After 24 Landrace male piglets (7.73 ± 0.38 kg) were fed for one week to adapt, they were randomly allocated into two groups according to their initial body weight, namely the control group (Control, *n* = 12) and the leptin treatment group (Leptin, *n* = 12). The leptin group was injected subcutaneously with 1 mg/kg leptin recombinant protein (ProSpec, Rehovot, Israel) for 4 weeks, and the control group was injected with equal-dose saline (PBS). Each piglet was fed ad libitum four times a day and had free access to water.

The body temperature of each piglet is measured daily. The mercury thermometer was inserted into the rectum of the piglet and was left for 3 min before taking it out to observe the temperature. Each piglet was repeatedly measured three times at a time, and the average value was taken as the day’s body temperature. At the end of the experiment, each piglet was slaughtered by electrocution. After slaughter, piglet liver, subcutaneous adipose tissue, longissimus dorsi, and omentum were collected and weighed immediately, frozen in liquid nitrogen, and transferred to −80 °C for storage until analysis. Three samples of 4-cm thickness were obtained at the longissimus muscle, minced, homogenized and a subsample of ~200 g was used for intramuscular fat content analysis. Intramuscular fat of all of the muscles was determined using near-IR FoodScan equipment (Foss Analytical, Hillerød, Denmark) according to the method described by Font-i-Furnols et al. [45]. The blood of 24 piglets was collected, and the separated serum was stored at −20 °C until analysis.

### 4.2. Primary Porcine Preadipocytes Culture and Induced Differentiation

According to the description of Du et al. [46], adipose tissues from the back of 3–5-day-old piglets were taken for Preadipocytes isolation and culture. Primary Preadipocytes were cultured in DMEM/F-12 (Gibco, 10565-018, New York City, NY, USA) supplemented with 10% FBS (Gibco, 10099–141), 2 mM L-glutamine, 100 μg/mL streptomycin, and 100 U/mL penicillin (Gibco, 15140–122) at 37 °C in 5% CO_2_. Differentiation is induced when the cells grow to 100%. Differentiation medium I is a basic medium containing 0.5 mmol/L IBMX (Sigma, St. Louis, MO, USA), 1 μmol/L dexamethasone (DEX, Sigma), and 5 mg/L insulin (Sigma). After 48 h, replace the basic culture medium (differentiation medium II) containing 5 mg/L insulin and culture for 48 h. After that, the medium was changed every 2 days until day eight.

### 4.3. Construction of Lipotoxicity Model

Primary Porcine Preadipocytes were induced to differentiate, and when the cells were differentiated to 8–12 days (80–90% showed adipocyte phenotype), they were used to establish the model. The FFA solution was prepared by the protein adsorption method [47], diluted with serum-free DMEM/F12 medium to concentrations of 0.1, 0.2, and 0.4 mmol/L, filtered and sterilized, and stored at −20 °C for 3 to 4 weeks. Palmitic acid was purchased from the Sigma company (St. Louis, MO, USA). Mature adipocytes were incubated with different concentrations of palmitic acid (0, 0.1, 0.2 and 0.4 mmol/L sodium palmitate +1% bull serum albumin + serum-free DMEM/12 medium) for 48 h.

### 4.4. m^6^A Methylation Agonists and Inhibitors

Cycloleucine (CL, Sigma, St. Louis, MO, USA) is used to inhibit the level of M^6^A methylation, and betaine (Bet, Sigma, St. Louis, MO, USA) is used to activate M^6^A methylation. CL treated porcine adipocytes at a concentration of 10 mM for 24 h, and Bet treated the cells at a concentration of 1 mM for 24 h.

### 4.5. Isolation, Purification and Identification of LD

The LD were isolated and purified according to the procedure described by Brasaemle et al. [48]. The collected pig fat cells are homogenized six to eight times with a tissue homogenizer. The homogenate was added to a buffer containing phenylmethylsulfonyl fluoride (PMSF, Sigma) and filtered through a 40-mesh cell sieve to obtain a cell slurry. The purification conditions were centrifugation at 230,000× *g* for 30 min at 4 °C. Coomassie brilliant blue staining identified rich and clear lipid droplet protein bands, and western blot was used to verify its purity.

### 4.6. FFA Content Analysis

FFA levels were measured using a commercially available kit (Nanjing Jiancheng Institute of Biological Engineering, Nanjing, China) in a microplate reader (PerkinElmer, Waltham, MA, USA) at wavelengths of 715 nm, respectively. The operating procedures were carried out according to the kit instructions. FFA (μmol/104 cell) = sample absorbance ÷ 0.0075 × 1 mL ÷ (1 mL÷1.2 mL × number of cells) = 0.16 × sample absorbance value ÷ number of cells.

### 4.7. Immunohistochemistry and Oil Red O Staining

Paraffin sections of adipose tissue were made and subjected to antigen retrieval and endogenous peroxidase activity blocking. Anti-Plin5 antibody (PA5-114352, Thermofisher, Waltham, MA, USA) was used as the primary antibody, and then the biotin-labeled secondary antibody (1:3000) was incubated for 1 h. The image was taken with an Olympus microscope.

After the frozen section (10 μm) of adipose tissue was made, the section was warmed up at room temperature and was immersed in distilled water. Then it was soaked in 60% isopropanol for 2 min, and was dyed with 5% oil red O working solution (Beyotime, Shanghai, China) for 5 min. It was toned with 60% isopropanol, and was washed and sealed with glycerin gelatin. The sections were observed under a microscope (Olympus, Tokyo, Japan).

### 4.8. Immunofluorescence and TUNEL Staining

The cell culture plate was washed three times with PBS, fixed with 4% paraformaldehyde for 15 min, and the fixative was discarded. The culture plate was incubated in PBS containing 0.3% Triton X-100 for 5 min. It was blocked in 10% goat serum blocking solution for 1 h. After overnight incubation with the anti-Plin5 antibody (Ab222811, Cambridge, UK) at 4 °C, the culture plate was incubated with the secondary antibody at 37 °C in the dark for 1 h. After DAPI staining for 5 min, the culture plate was observed and photographed under a fluorescence microscope (BioTek, Winooski, VT, USA). The preliminary operation of TUNEL staining is the same as that for immunofluorescence. The culture plate was incubated in 0.3% Triton X-100 PBS for 5 min, and then TUNEL staining solution (Vazyme, Nanjing, China) was added to the plate and incubated for 1 h at 37 °C in the dark.

### 4.9. BODIPY Dyeing

BODIPY staining solution was purchased from Beyotime (Shanghai, China). The cells fixed with 4% paraformaldehyde were stained with BODIPY staining working solution for 30 min, and then DAPI stained for 10 min. The results were observed using a fluorescence microscope (BioTek, Winooski, VT, USA).

### 4.10. Dot Blot Analysis

Total cell RNA was extracted and diluted to a concentration of 50 ng/μL. The extracted RNA was heated at 95 °C for 3 min, and 2 μL was dropped evenly on the nitrocellulose (NC) membrane. The membrane was cross-linked by UV with 1×PBST (phosphate buffer saline (PBS), 0.05% Tween 20). The membrane was incubated with an anti-m^6^A antibody (1:100, ABCAM, Cambridge, UK) for 7–10 h at 4 °C, and then incubated with HRP-conjugated goat anti-mouse IgG (Boster, Wuhan, China) for 1 h at room temperature. After the developer (YEASEN, Shanghai, China) was dropped on the NC membrane and left to stand for 2 min, the membrane was placed on the BIO-RAD chemiluminescence gel imaging system (Bio-Rad, Hercules, CA, USA) to observe and save the pictures.

### 4.11. m^6^A-IP

The total RNA in the adipocytes was extracted, and the mRNA was isolated and purified using PolyA Ttract mRNA Isolation Systems (promega, Madison, WI, USA). The purified mRNA was added to mRNA denaturation buffer (10 mM ZnCl2, 10 mM Tris-HCl, pH = 7.0) and was heated at 94 °C for 5 min. The mRNA was further added to the buffer (150 mM NaCl, 0.1% NP-40, 10 mM Tris-HCl, pH = 7.0) and was immunoprecipitated with anti-m^6^A antibody (ABCAM, Cambridge, UK) for 2 h at 4 °C. Dynabeads Protein G (Thermo Fisher, Waltham, Massachusetts, USA) was added to the mixture and reacted at 4 °C for 2 h. After the m^6^A-positive mRNA was eluted, the mRNA was recovered with ethanol and analyzed by real-time quantitative PCR.

### 4.12. m^6^A Quantification by Colorimetric Assay

The total M^6^A methylation level was determined as described in our previous study [49]. That is, the total RNA is extracted from the adipocytes and the mRNA is purified using the PolyATtracre mRNA separation system (Promega, Madison, WI, USA). Then, the total RNA M^6^A methylation level was analyzed by the M^6^A RNA methylation quantification kit (EpiQuik, New York, NY, USA).

### 4.13. Plasmid Construction and RNA Interference

The FTO overexpression vector in this study was constructed from previous studies [50]. The FTO interference fragment was constructed in Gene Pharma (Shanghai, China). Overexpression plasmid vectors of Plin5 (pc-Plin5) were constructed in our lab (Forward: 5′ -CCCAAGCTTATGCATCATCACCATCACCATTCAGAAGAAGAGGGGGCTCA -3′, reverse: 5′ -CCGCTCGAGTCAAAAGTCCAGCTCTGGCATCA-3′; Hind Ⅲ: AAGCTT, Xho I: CTCGAG, His Tag: CATCATCACCATCACCAT). The Plin5 gene sequence was cloned into the pcDNA3.1 vector (Invitrogen) by standardized procedures. Cells were transfected at 70% confluence using X-tremeGENE HP DNA Transfection Reagent (Roche, Carlsbad, CA, USA). All plasmid transfection procedures were performed in accordance with the manufacturer’s instructions.

### 4.14. Luciferase Reporter Assay

The sequence containing the wild-type and mutant Plin5–3′ UTR was subcloned into pGL3-basic luciferase vector (Takara, Japan). Both the wild-type and Plin5 3′UTR mutant vectors were transfected into the differentiated porcine preadipocytes. After 48 h of transfection, the cells were collected and analyzed for activity using the Dual-Luciferase Reporter assay system (Promega, Madison, WI, USA).

### 4.15. qPCR and Western Blotting Analysis

TRIzol reagent (Invitrogen, Thermo Fisher Scientific, Carlsbad, CA, USA) was used to extract total RNA according to the manufacturer’ s instructions. The mRNA expression of related genes was analyzed by real-time PCR. Briefly, complementary DNA (cDNA) was synthesized using the PrimeScript RT reagent kit (TaKaRa Bio, Inc., Otsu, Japan). The SYBR Premix Ex Taq II kit (TaKaRa) was used for real-time PCR in an ABI Step One System (Applied Biosystems, Foster City, CA, USA). The reaction program was set to 95 °C, 10 s (pre-change period), 95 °C, 5 s, 60 °C, 30 s (PCR reaction period), a total of 40 cycles. The GAPDH gene is used to normalize the relative expression of the target gene according to the formula 2^−ΔΔCt^. All primer sequences are shown in Appendix A.

Western blotting analysis was performed using the standard method described by Liu et al. [31]. Perilipin5 (Ab222811, Cambridge, UK), Perilipin2 (ab78920, Cambridge, UK), Perilipin1 (ab172907, Cambridge, UK), ATGL (ab109251, Cambridge, UK), PPARγ (ab59256, Cambridge, UK), HSL (ab45422, Cambridge, UK), CPT1a (ab220789, Cambridge, UK), PPARGC1A(ab191838, Cambridge, UK), NDUFB8 (ab110242, Cambridge, UK), DGAT2 (ab237613, Cambridge, UK) and LPL (ab93898, Cambridge, UK) are all purchased from Abcam. β-actin (AP0060) and GAPDH (AP0063) were purchased from Bioworld (Nanjing, China). HRP-labeled goat anti-rabbit IgG (WLA023) was purchased from Wanleibio (Shenyang, China). The gray value of each band was analyzed by ImageLab software and was normalized by β-actin or GAPDH to calculate the relative expression of the target protein.

### 4.16. NAD^+^/ NADH Analysis

The analysis of NAD^+^/NADH was performed using an NAD/NADH Assay Kit (Colorimetric, ab65348, Cambridge, UK). Cell/tissue samples were extracted with extraction buffer and were deproteinized with spin columns. Samples and standards were added to the wells, and the reaction mixture was added and incubated at room temperature for 5 min to convert NAD to NADH. Then each well was added with NADH chromogenic reagent and incubated during the reaction cycle for 1–4 h. During the 1–4 h incubation period, the microplate reader was used several times to analyze the absorbance value. The reaction can be terminated with a stop solution. The NAD/NADH Ratio was calculated as: NAD/NADH ratio = (NAD−NADH)/NADH.

### 4.17. Statistical Analysis

All data were preliminarily organized using Excel 2016. Kolmogorov–Smirnov tests were used to test the data for normality in the SAS9.4 software (SAS Inst. Inc., Cary, NC, USA). The control and leptin-treated data were analyzed by independent samples *T* test for significance. The other in vitro cell treatment experiments of three groups or more were analyzed by one-way ANOVA. The data are presented as means ± standard deviation (Means ± SD). *p* < 0.01 ** is considered to be extremely significant, *p* < 0.05 * is considered significant.

## 5. Conclusions

This study found that leptin up-regulated the expression of Plin5 to promote mitochondrial function and lipolysis, further verifying that the up-regulation of Plin5 is modified by FTO-mediated M^6^A methylation.

## Figures and Tables

**Figure 1 ijms-22-10610-f001:**
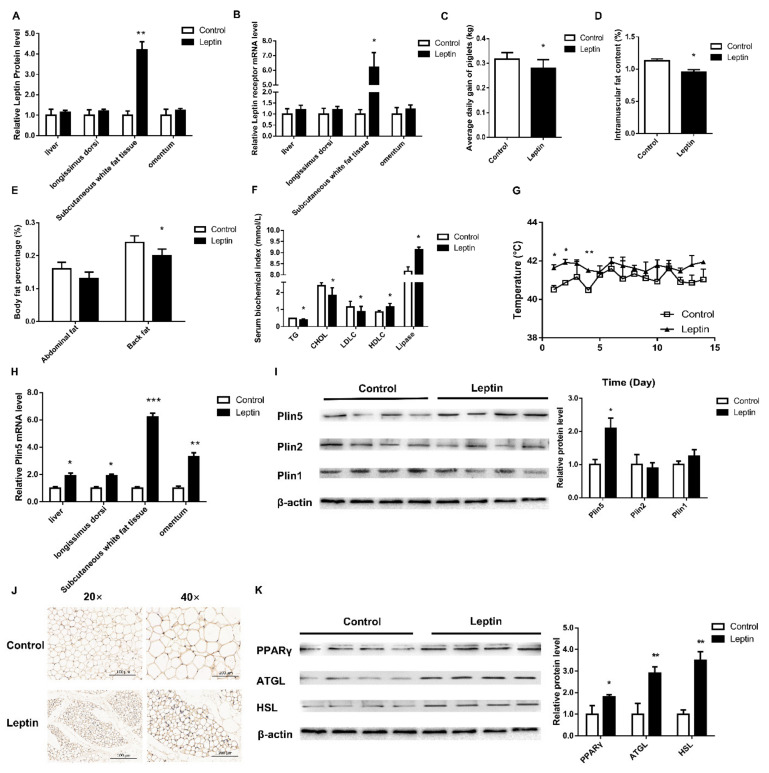
Leptin improves the body fat composition and weight of piglets by up-regulating the expression of Plin5. (**A**) The leptin kit was used to detect the level of leptin protein in various tissues of piglets. (**B**) The expression of leptin receptors in various tissues of piglets. The effect of leptin treatment on: (**C**) the average daily gain, (**D**) the intramuscular fat content, (**E**) abdominal fat rate and back fat rate and (**F**) blood lipid level etc. (**G**) The change of piglet’s body temperature for 15 days after leptin treatment. (**H**,**I**) The mRNA and protein expression levels of the Plin family were analyzed by real-time quantitative PCR and western blot. (**J**) Immunohistochemical detection of piglet adipose tissue. (**K**) Western blot analysis of lipolytic protein expression. *n* = 4 in each group, values are means ± SD. vs. control group, * *p* < 0.05, ** *p* < 0.01, *** *p* < 0.001.

**Figure 2 ijms-22-10610-f002:**
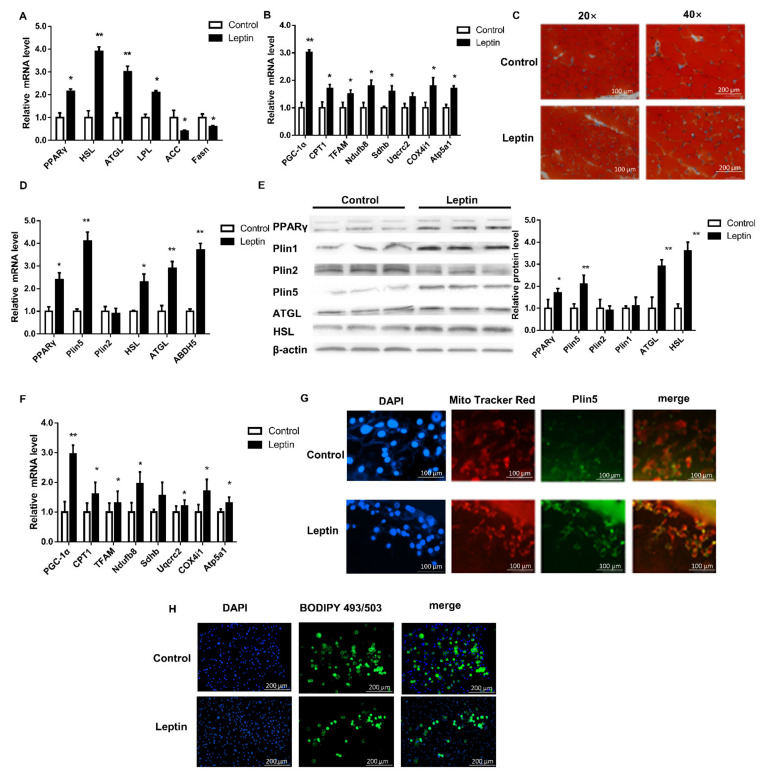
Leptin promotes lipolysis by up-regulating the expression of Plin5 in pig adipocytes. (**A**,**B**) Real-time quantitative PCR was used to detect the mRNA expression of Plin5, lipolytic genes and mitochondrial complex-related genes in piglets subcutaneous adipose tissue. *n* = 4 in each group. (**C**) Oil red O staining was used to observe the number and size of LD in the subcutaneous fat tissue of piglets. (**D**) Real-time quantitative PCR was used to detect Plin5 and lipolytic gene expression in porcine adipocytes in vitro treated with 50 nmol/μL leptin for 24 h. The effect of 50 nmol/μL leptin treatment of porcine adipocytes in vitro on: (**E**) the expression of Plin5 and lipolytic protein and (**F**) the mRNA expression of mitochondrial complex-related genes. (**G**,**H**) Porcine adipocytes in vitro were treated with 50 nmol/μL leptin for 24 h to detect Plin5 immunofluorescence or BODIPY staining analysis. *n* = 3 for each group of cell samples, values are means ± SD. vs. control group, * *p* < 0.05, ** *p* < 0.01.

**Figure 3 ijms-22-10610-f003:**
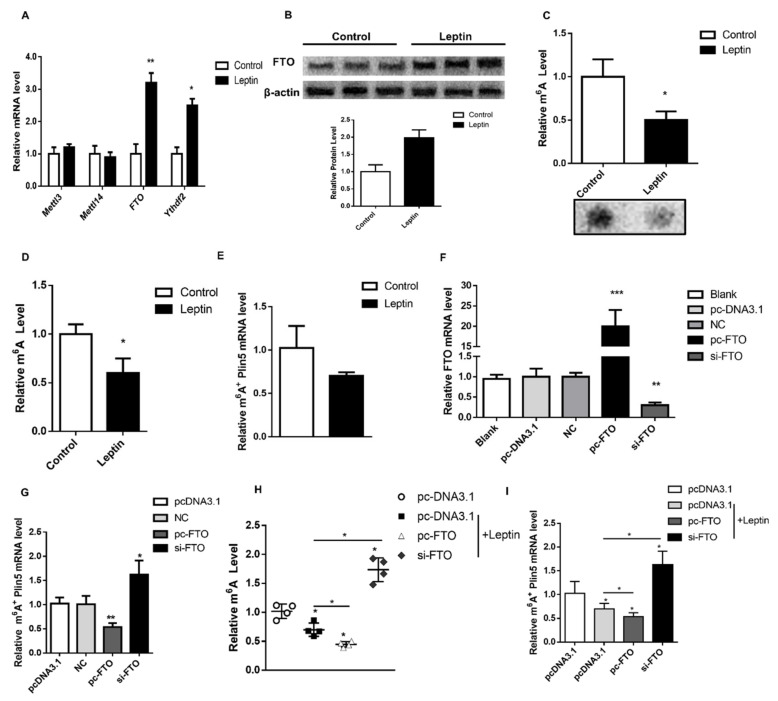
Leptin inhibits the m^6^A methylation of Plin5 through FTO. (**A**) Porcine adipocytes in vitro were treated with 50 nmol/μL leptin for 24 h to detect the mRNA expression of methylation-related genes. (**B**) The protein expression of FTO under the same treatment as (**A**). (**C**) Dot Blot detected the level of total RNA m^6^A in 50 nmol/μL leptin-treated porcine adipocytes in vitro. (**D**) The colorimetric assay was used to quantify the relative m^6^A levels of 50 nmol/μL leptin-treated porcine adipocytes in vitro. (**E**) m^6^A-IP was used to detect Plin5 M^6^A methylation in 50 nmol/μL leptin-treated porcine adipocytes in vitro. (**F**) Real-time quantitative PCR was used to detect FTO overexpression and interference efficiency in vitro. (**G**) m^6^A-IP detected the m^6^A level of Plin5 after overexpression or interference with FTO in vitro. (**H**) The relative M^6^A level of 50 nmol/μL leptin treated porcine adipocytes in vitro after overexpression or interference with FTO. (**I**) Plin5 M^6^A levels in 50 nmol/μL leptin-treated porcine adipocytes in vitro after overexpression or interference with FTO. *n* = 6 for each group of cell samples, values are means ± SD. vs. control group, * *p* < 0.05, ** *p* < 0.01, *** *p* < 0.001.

**Figure 4 ijms-22-10610-f004:**
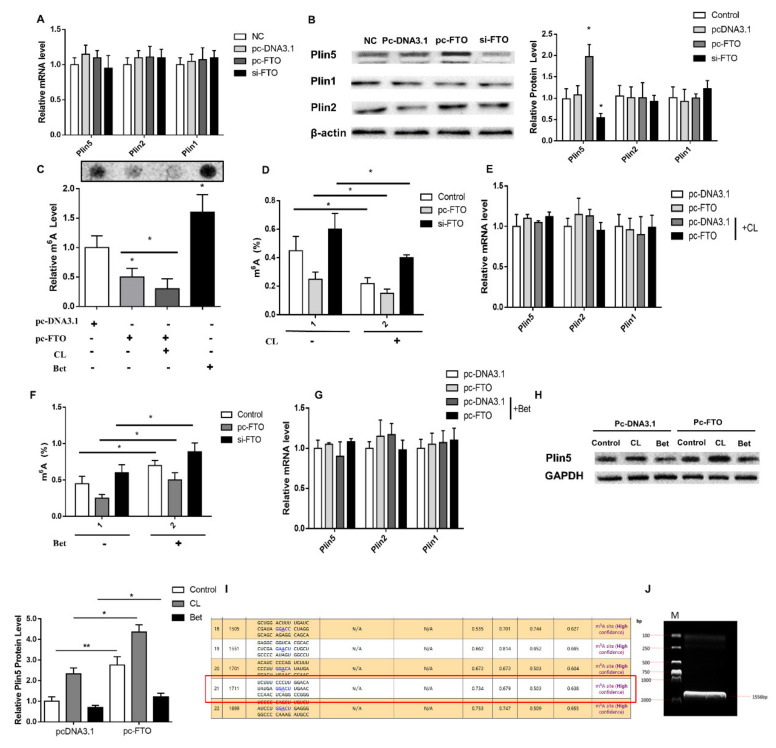
FTO up-regulates Plin5 protein expression by inhibiting the m^6^A methylation of Plin5. The effect of FTO interference or overexpression in vitro on: (**A**) Plin family mRNA expression and (**B**) Plin family protein levels. (**C**) The relative M^6^A levels of pig adipocytes in vitro treated with CL and Bet after overexpression or interference with FTO. (**D**) The total M^6^A level of CL−treated pig adipocytes in vitro after overexpression or interference with FTO. (**E**) Plin family mRNA expression in CL−treated pig adipocytes in vitro after overexpression or interference with FTO. (**F**) The total M^6^A level of Bet-treated pig adipocytes after overexpression or interference with FTO. (**G**) Plin family mRNA expression in Bet-treated pig adipocytes in vitro after overexpression FTO. (**H**) Plin5 protein expression in pig adipocytes in vitro treated with CL and Bet after overexpression FTO. (**I**,**J**) Plin5 3′UTR end M^6^A site mutation vector construction. *n* = 4 for each group of cell samples, values are means ± SD. vs. control group, * *p* < 0.05, ** *p* < 0.01.

**Figure 5 ijms-22-10610-f005:**
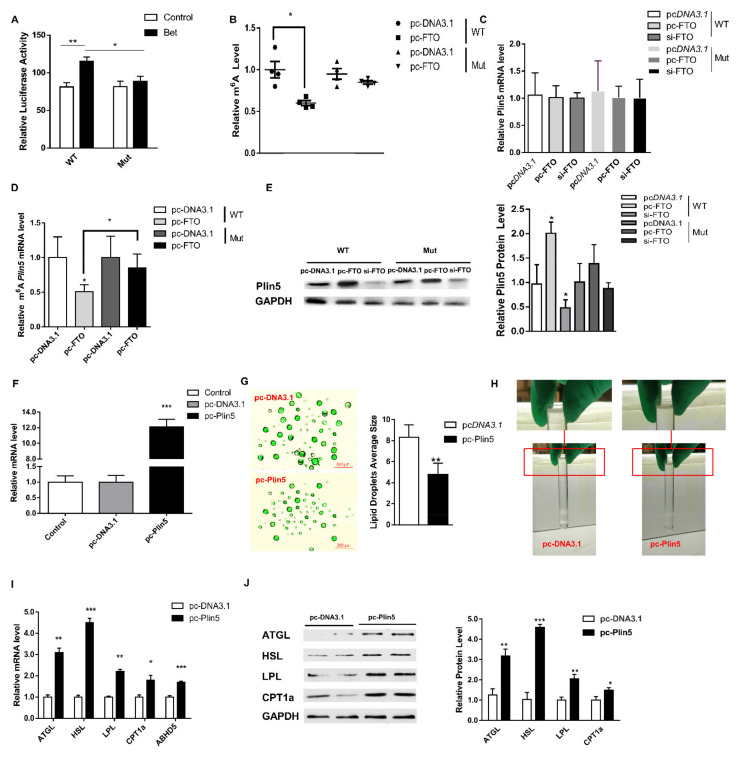
Plin5 regulates lipid metabolism and energy metabolism through LD. (**A**) Luciferase reporter assay was used to analyze the effect of M^6^A mutation at the 3′UTR end of Plin5. (**B**) Relative M^6^A level analysis of wild-type and Plin5 mutant. (**C**) Detection of Plin5 mRNA expression of wild type and Plin5 mutant type. (**D**) Plin5 m^6^A methylation analysis of wild type and Plin5 mutant. (**E**) Plin5 protein expression of wild type and Plin5 mutant type. (**F**) Real-time quantitative PCR was used to detect the efficiency of the Plin5 overexpression vector. (**G**) Porcine adipocytes in vitro were transfected with Plin5 overexpression vector and then subjected to BODIPY fluorescent staining of LD. (**H**) Observation of lipid droplet extraction. (**I**,**J**) Porcine adipocytes in vitro were transfected with Plin5 over-expression vector to detect lipolysis-related genes mRNA or protein levels. *n* = 4 for each group of cell samples, values are means ± SD. vs. control group, * *p* < 0.05, ** *p* < 0.01, *** *p* < 0.001.

**Figure 6 ijms-22-10610-f006:**
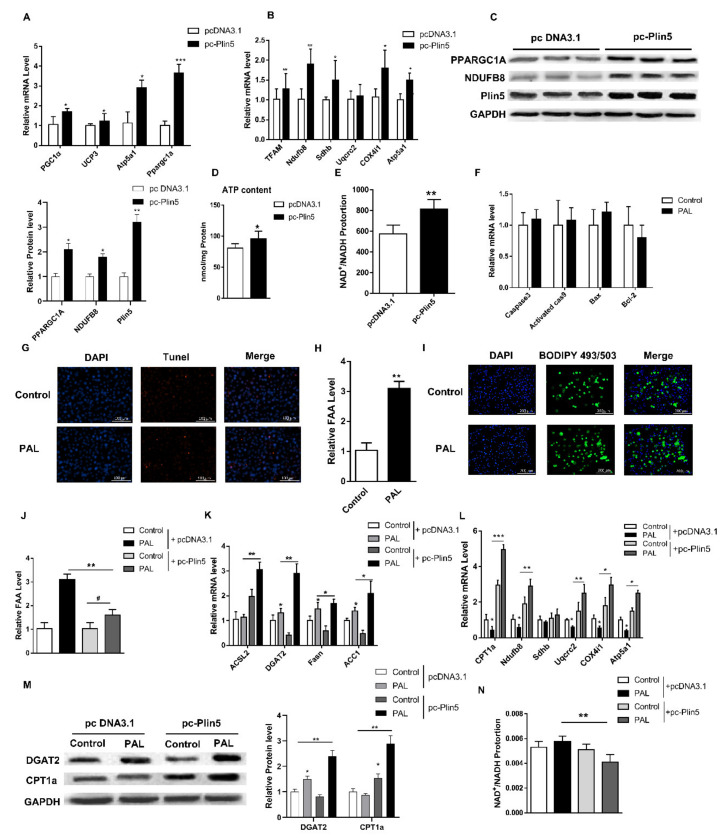
Plin5 promotes lipid synthesis and enhances mitochondrial β-oxidation in a lipotoxic model. (**A**,**B**) Real-time quantitative PCR was used to detect mitochondrial functional gene and β-oxidation gene mRNA expression after Plin5 overexpression. (**C**) Under the same treatment as (**A**,**B**), the protein levels of differentially expressed genes were analyzed by Western blotting. (**D**) The kit detected the ATP content of porcine adipocytes transfected with Plin5 overexpression vector. (**E**) The ratio of NAD^+^/NADH was measured after Plin5 overexpression in porcine adipocytes in vitro. (**F**) 0.4 mmol/L PAL treated porcine adipocytes in vitro for 48 h to detect the mRNA expression of apoptosis-related genes. (**G**) Porcine adipocytes in vitro were treated with 0.4 mmol/L PAL for 48h to perform Tunel staining. (**H**) The FFA level was detected by the kit after treatment of 0.4 mmol/L PAL with porcine adipocytes in vitro for 48 h. (**I**) BODIPY staining after the same treatment in vitro. (**J**) After Plin5 overexpression, 0.4 mmol/L PAL was used to treat porcine adipocytes in vitro for 48 h to detect FFA levels. (**K**,**L**) After Plin5 was overexpressed, pig adipocytes in vitro were treated with 0.4 mmol/L PAL for 48 h to detect the mRNA levels of lipid synthesis genes and mitochondrial β-oxidation genes. (**M**) Under the same treatment as (**K**,**L**), the protein levels of GDAT2 and CPT1a were analyzed by Western blotting. (**N**) The NAD^+^/NADH ratio is measured after the same treatment as (**J**). *n* = 4 for each group of cell samples, values are means ± SD. vs. control group, * *p* < 0.05, ** *p* < 0.01, *** *p* < 0.001.

## Data Availability

The data presented in this study are available on request from the corresponding author.

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
