# Peer review of "Leptin Reduces Plin5 m6A Methylation through FTO to Regulate Lipolysis in Piglets"

_ijms, 2021, doi:10.3390/ijms221910610_

Round 1

Reviewer 1 Report

Summary: The authors have studied the effect of leptin on Plin5 m6A mRNA methylation in adipose tissue. It was shown that leptin increased FTO which decreased Plin5 m6A methylation and increased Plin5 protein. Plin5 reduced lipid droplet size, increased TG metabolism and promoted thermogenesis. The treatment of piglets with leptin for 4 weeks was a good in vivo model (Fig 1). Leptin treatment increased Plin5 protein and mRNA. This observation was confirmed in pig adipocytes (Fig. 2). The data in fig 4 identify differences in leptin and FTO regulation of Plin5 mRNA as FTO does not regulate Plin5 mRNA abundance but does alter m6A methylation.

Comments:

This is a solid mechanistic study that provides new insights into the regulation of adipose tissue metabolism by leptin via Plin5.

  1. Figure 6 was not included in the manuscript. These data are discussed in the results on pg 9 of the manuscript.

  1. Leptin increases FTO and Plin5 mRNA. However, FTO does not alter Plin5 mRNA abundance just the m6A methylation status. The authors should provide a brief comment in the discussion about potential mechanisms for leptin’s effect on Plin5 mRNA as it appears not to be through FTO. It seems that leptin may have 2 mechanisms for elevating Plin5 protein – elevated mRNA abundance and decreased m6A methylation via FTO?

Minor comments:

Panel 1A: Why did the treatment with leptin increase leptin mRNA in adipose tissue? Is this a known effect?

Fig 1: There are some additional comments about panels D and E at the bottom of figure legend 1 that need to be removed.

Panel 2E: Leptin is misspelled. Also in figure 1, “leptine” is used rather than “leptin” in some panels.

Fig 2: It should be mentioned which fat pad was used to make adipocytes? In the methods, three fat sources are noted “neck, scarpula and back.” Are these cells equivalent? This reviewer is not familiar with the piglet, but at least in mice some fat pads are much more prone to beiging or being associated with BAT.

Fig 3A: The rationale for measuring Ythdf2 should be mentioned. It does not come up again in the manuscript.

Fig 3: Leptin increases FTO mRNA. The protein levels of FTO are not shown. Can the authors provide a Western showing the effect of leptin on FTO protein?

Author Response

Dear reviewer,

We would like to thank you very much for your valuable comments and good suggestions that greatly helped to improve our manuscript. Thank you very much for your time and efforts. We have studied comments carefully and have made correction which we hope meet with approval. At the same time, we also invited a native English-speaking colleague to revise the grammar of the article and make the appropriate corrections. Revised portion are highlighted in the track changes mode in MS Word. We hope that our revision can meet your requirements. The main corrections in the paper and the responds to the reviewer’s comments are as flowing:

Point 1: Figure 6 was not included in the manuscript. These data are discussed in the results on pg 9 of the manuscript.

Response 1: Thank you for your careful review. There are a total of 7 figures in our manuscript, but the last two figures in the PDF version of the manuscript are not displayed properly. We will contact the editor to deal with the situation. We are very sorry for the trouble to you.

Point 2: Leptin increases FTO and Plin5 mRNA. However, FTO does not alter Plin5 mRNA abundance just the m6A methylation status. The authors should provide a brief comment in the discussion about potential mechanisms for leptin’s effect on Plin5 mRNA as it appears not to be through FTO. It seems that leptin may have 2 mechanisms for elevating Plin5 protein – elevated mRNA abundance and decreased m6A methylation via FTO?

Response 2: We feel great thanks for your professional suggestion on our manuscript. As you mentioned, our data shows that leptin treatment promotes PLin5 mRNA levels in piglets, but in vitro FTO overexpression has no effect on Plin5 mRNA levels. We did not mention the potential mechanism of factors other than FTO in regulating the abundance of Plin5 mRNA. We speculate that the changes of Plin5 mRNA are related to the regulation of lipolysis-related genes by leptin. Because it is shown in our data that leptin treatment has increased the mRNA and protein expression of PPARγ, HSL, ATGL, and LPL. Other reports have found that PPARγ induces the accumulation of Plin5 [1], and further HSL and ATGL can interact with Plin5 to regulate lipolysis [2-4]. Our laboratory previously reported that leptin regulates the JAK2/STAT3 signaling pathway through SOCS3, leading to the up-regulation of Plin5 expression [5]. We have also added the potential mechanism of leptin other than FTO to regulate Plin5 mRNA in the discussion. Please refer to the revised manuscript for details.

Point 3: Panel 1A: Why did the treatment with leptin increase leptin mRNA in adipose tissue? Is this a known effect?

Response 3: We are very sorry for the wrong labeling of the ordinate in Figure 1A. Figure 1A detects relative leptin protein levels in various tissues, not relative leptin mRNA expression. We have modified the corresponding coordinate labeling. For details, please refer to Figure 1A in the revised manuscript. Thank you for your reminder.

Point 4: Fig 1: There are some additional comments about panels D and E at the bottom of figure legend 1 that need to be removed.

Response 4: Thank you for your reminding. We have modified the phrase "Effect of leptin treatment on..." in Figure 1D and E to "Effect of leptin treatment on: (C)..., (D)..., (E)...etc."

Point 5: Panel 2E: Leptin is misspelled. Also in figure 1, “leptine” is used rather than “leptin” in some panels.

Response 5: We are very sorry for the misspelling of leptin. We carefully checked all the spellings in the entire manuscript and pictures and made the correct corrections accordingly.

Point 6: Fig 2: It should be mentioned which fat pad was used to make adipocytes? In the methods, three fat sources are noted “neck, scarpula and back.” Are these cells equivalent? This reviewer is not familiar with the piglet, but at least in mice some fat pads are much more prone to beiging or being associated with BAT.

Response 6: Thank you very much for your valuable suggestions in our manuscript. It is reported that pigs have no brown adipose tissue and lack a functional UCP1 gene [6]. Therefore, the three sources of adipose tissue are equivalent when making adipocytes in pigs, all of which are white adipose tissue. Because we are worried that the piglets of 3-5 days old will have too little fat pads, we took multiple parts of white adipose tissue for isolation and culture. But the Primary Porcine Preadipocytes used in the final experiment came from the back.

Point 7: Fig 3A: The rationale for measuring Ythdf2 should be mentioned. It does not come up again in the manuscript.

Response 7: Thank you for your valuable suggestions to improve the quality of our manuscript. Ythdf2 specifically recognizes and binds RNA containing N6-methyladenosine (M6A) and regulates its stability [7-9]. Therefore, we evaluated it as a key gene regulating M6A methylation. We have also made corresponding supplements to the rationale of detecting it in the results part.

Point 8: Fig 3: Leptin increases FTO mRNA. The protein levels of FTO are not shown. Can the authors provide a Western showing the effect of leptin on FTO protein?

Response 8: Thank you for your careful review. We have supplemented the FTO protein level in Figure 3B. As shown in the figure, leptin treatment significantly increased FTO protein expression.

References

  1. Tian, S.; Lei, P.; Teng, C.; Sun, Y.; Song, X.; Li, B.; Shan, Y., Targeting PLIN2/PLIN5-PPARγ: Sulforaphane Disturbs the Maturation of Lipid Droplets. Molecular nutrition & food research 2019, 63, (20), e1900183.
  2. Gallardo-Montejano, V. I.; Yang, C.; Hahner, L.; McAfee, J. L.; Johnson, J. A.; Holland, W. L.; Fernandez-Valdivia, R.; Bickel, P. E., Perilipin 5 links mitochondrial uncoupled respiration in brown fat to healthy white fat remodeling and systemic glucose tolerance. Nature communications 2021, 12, (1), 3320-3320.
  3. Whytock, K. L.; Shepherd, S. O.; Wagenmakers, A. J. M.; Strauss, J. A., Hormone-sensitive lipase preferentially redistributes to lipid droplets associated with perilipin-5 in human skeletal muscle during moderate-intensity exercise. J Physiol 2018, 596, (11), 2077-2090.
  4. Granneman, J. G.; Moore, H.-P. H.; Mottillo, E. P.; Zhu, Z.; Zhou, L., Interactions of perilipin-5 (Plin5) with adipose triglyceride lipase. The Journal of biological chemistry 2011, 286, (7), 5126-5135.
  5. Liu, Z.; Gan, L.; Zhou, Z.; Jin, W.; Sun, C., SOCS3 promotes inflammation and apoptosis via inhibiting JAK2/STAT3 signaling pathway in 3T3-L1 adipocyte. Immunobiology 2015, 220, (8), 947-53.
  6. Zheng, Q.; Lin, J.; Huang, J.; Zhang, H.; Zhang, R.; Zhang, X.; Cao, C.; Hambly, C.; Qin, G.; Yao, J.; Song, R.; Jia, Q.; Wang, X.; Li, Y.; Zhang, N.; Piao, Z.; Ye, R.; Speakman, J. R.; Wang, H.; Zhou, Q.; Wang, Y.; Jin, W.; Zhao, J., Reconstitution of UCP1 using CRISPR/Cas9 in the white adipose tissue of pigs decreases fat deposition and improves thermogenic capacity. Proc. Natl. Acad. Sci. U. S. A. 2017, 114, (45), E9474-e9482.
  7. Wang, X.; Lu, Z.; Gomez, A.; Hon, G. C.; Yue, Y.; Han, D.; Fu, Y.; Parisien, M.; Dai, Q.; Jia, G.; Ren, B.; Pan, T.; He, C., N6-methyladenosine-dependent regulation of messenger RNA stability. Nature 2014, 505, (7481), 117-20.
  8. Wang, X.; Zhao, B. S.; Roundtree, I. A.; Lu, Z.; Han, D.; Ma, H.; Weng, X.; Chen, K.; Shi, H.; He, C., N(6)-methyladenosine Modulates Messenger RNA Translation Efficiency. Cell 2015, 161, (6), 1388-99.
  9. Xu, C.; Liu, K.; Ahmed, H.; Loppnau, P.; Schapira, M.; Min, J., Structural Basis for the Discriminative Recognition of N6-Methyladenosine RNA by the Human YT521-B Homology Domain Family of Proteins. The Journal of biological chemistry 2015, 290, (41), 24902-13.

Reviewer 2 Report

GENERAL COMMENTS

The manuscript addresses a topic of interest as well as applicability in the clinical setting, which is within the journal’s scope. A huge amount of information is summarized by the authors derived from multiple experiments. The structure of the manuscript makes it easy to follow and read. The figures are welcome to illustrate the information presented.

Some recommendations are provided below in order to improve the manuscript:

Title and Abstract: it should be mentioned that the study was performed in piglets. In the abstract no information about the experimental groups/design is provided.

Introduction, page 1, line 28 and throughout the whole manuscript: use people first language, so that “obese individuals/subjects” is replaced by “individuals/subjects with obesity”.

Introduction, page 1, line 29: suggest to replace reference 1 by a more recent and updated review like Bakshi A, Singh R, Rai U. Trajectory of leptin and leptin receptor in vertebrates: Structure, function and their regulation. Comp Biochem Physiol B Biochem Mol Biol. 2021 Jul 31;257:110652.

Introduction, page 2, last paragraph: formulate a hypothesis (in page 8, line 198 it is mentioned that the findings “support our hypothesis”, but it was not explicitly  introduced).

Discussion, page 9, lines 252-253: I am afraid that I do not understand the sentence that “…leptin can also promote body fat consumption in pigs”; please eliminate or rephrase.

Discussion: A clear understanding of the link between FTO intronic variants and FTO activity has remained elusive. It has been shown that obesity-associated SNPs appear functionally connected not with FTO but with two neighboring genes: IRX3 and RPGRIP1L (ref Tung YCL, Yeo GSH, O'Rahilly S, Coll AP. Obesity and FTO: Changing Focus at a Complex Locus. Cell Metab. 2014 Nov 4;20(5):710-718. doi: 10.1016/j.cmet.2014.09.010. Epub 2014 Oct 23. PMID: 25448700). This should be mentioned in the Discussion.

Discussion: type 2 diabetes (T2D) is a frequent comorbidity in patients with obesity. It is worth emphasizing that obesity and T2D are diseases that need a combined treatment (ref Leitner DR et al. Obesity and Type 2 Diabetes: Two Diseases with a Need for Combined Treatment Strategies - EASO Can Lead the Way. Obes Facts. 2017;10(5):483-492).

Discussion: the authors focus their findings only on the effect of leptin on plin5 methylation via FTO to regulate lipolysis. However, leptin is well known to exert other effects on lipolysis that should at least also mentioned (ref Frühbeck G, Gómez-Ambrosi J, Salvador J. Leptin-induced lipolysis opposes the tonic inhibition of endogenous adenosine in white adipocytes. FASEB J. 2001 Feb;15(2):333-40  // Frühbeck G, Gómez-Ambrosi J. Modulation of the leptin-induced white adipose tissue lipolysis by nitric oxide. Cell Signal. 2001 Nov;13(11):827-33). Moreover, in relation to the role of glucose tolerance and dysfunctional adiposity, it would be worthwhile mentioning the relevance of the leptin-adiponectin ratio (ref  Frühbeck G, Catalán V, Rodríguez A, Ramírez B, Becerril S, Salvador J, Portincasa P, Colina I, Gómez-Ambrosi J. Involvement of the leptin-adiponectin axis in inflammation and oxidative stress in the metabolic syndrome. Sci Rep. 2017 Jul 26;7(1):6619).

Discussion: furthermore, as regards body composition and effect on body temperature it is also important to mention the effect of leptin on myogenesis (ref Rodríguez A, Becerril S, Méndez-Giménez L, et al. Leptin administration activates irisin-induced myogenesis via nitric oxide-dependent mechanisms, but reduces its effect on subcutaneous fat browning in mice. Int J Obes (Lond). 2015 Mar;39(3):397-407).

Mat & Methods: the methodology is adequately described.

Author Response

Dear reviewer,

We would like to thank you very much for your valuable comments and good suggestions that greatly helped to improve our manuscript. Thank you very much for your time and efforts. We have studied comments carefully and have made correction which we hope meet with approval. At the same time, we also invited a native English-speaking colleague to revise the grammar of the article and make the appropriate corrections. Revised portion are highlighted in the track changes mode in MS Word. We hope that our revision can meet your requirements. The main corrections in the paper and the responds to the reviewer’s comments are as flowing:

Point 1: Title and Abstract: it should be mentioned that the study was performed in piglets. In the abstract no information about the experimental groups/design is provided.

Response 1: Thank you for your careful review. We have added the content of the piglet-related trial groups/design in the title and abstract. Please refer to the revised manuscript for details.

Point 2: Introduction, page 1, line 28 and throughout the whole manuscript: use people first language, so that “obese individuals/subjects” is replaced by “individuals/subjects with obesity”.

Response 2: Thank you for your reminding. We are very sorry for the language error in our manuscript. We carefully checked the whole manuscript and revised them. The revised paragraphs include lines 30-31 and 36 on page 2, lines 75-76 on page 4, line 433 on page 18.

Point 3: Introduction, page 1, line 29: suggest to replace reference 1 by a more recent and updated review like Bakshi A, Singh R, Rai U. Trajectory of leptin and leptin receptor in vertebrates: Structure, function and their regulation. Comp Biochem Physiol B Biochem Mol Biol. 2021 Jul 31;257:110652.

Response 3: Thank you for your valuable suggestions to improve the quality of our manuscript. We have replaced reference 1 with the recent review by Bakshi A et al [1].

Point 4: Introduction, page 2, last paragraph: formulate a hypothesis (in page 8, line 198 it is mentioned that the findings “support our hypothesis”, but it was not explicitly  introduced).

Response 4: Thank you for your reminding. Your suggestions are very valuable for perfecting our manuscript, and we have formulated a hypothesis in the penultimate and last paragraph of the introduction. Please refer to the revised manuscript for details.

Point 5: Discussion, page 9, lines 252-253: I am afraid that I do not understand the sentence that “…leptin can also promote body fat consumption in pigs”; please eliminate or rephrase.

Response 5: Thank you for your careful review. We are very sorry that the expression of this sentence is not clear. In the discussion we have deleted it.

Point 6: Discussion: A clear understanding of the link between FTO intronic variants and FTO activity has remained elusive. It has been shown that obesity-associated SNPs appear functionally connected not with FTO but with two neighboring genes: IRX3 and RPGRIP1L (ref Tung YCL, Yeo GSH, O'Rahilly S, Coll AP. Obesity and FTO: Changing Focus at a Complex Locus. Cell Metab. 2014 Nov 4;20(5):710-718. doi: 10.1016/j.cmet.2014.09.010. Epub 2014 Oct 23. PMID: 25448700). This should be mentioned in the Discussion.

Response 6: We are very grateful for your appropriate suggestion and are willing to accept it. As you said, FTO has a role in controlling energy homeostasis and body composition, but there is still much to learn. For example, in humans, the expression changes of RPGRIP1L are related to FTO, and the SNP related to obesity does not seem to be related to FTO but RPGRIP1L and IRX3 functionally [2]. This elusive link between FTO intron variation and FTO activity is indeed not discussed in the manuscript. However, these reports are closely related to our research, so we have supplemented the corresponding content in the discussion. Please refer to the revised manuscript for details. At the same time, these interesting reports also provide new directions for future research in our laboratory.

Point 7: Discussion: type 2 diabetes (T2D) is a frequent comorbidity in patients with obesity. It is worth emphasizing that obesity and T2D are diseases that need a combined treatment (ref Leitner DR et al. Obesity and Type 2 Diabetes: Two Diseases with a Need for Combined Treatment Strategies - EASO Can Lead the Way. Obes Facts. 2017;10(5):483-492).

Response 7: Thank you for your careful review. We have carefully read the references provided by you and have further supplemented them in the discussion. Please refer to the revised manuscript for details. Obesity is indeed closely related to T2D, and anti-obesity is often used as one of the treatment strategies for T2D in clinical practice. The data of this study provides a new strategy for weight loss and body fat control, which also brings a new direction for the treatment of T2D.

Point 8: Discussion: the authors focus their findings only on the effect of leptin on plin5 methylation via FTO to regulate lipolysis. However, leptin is well known to exert other effects on lipolysis that should at least also mentioned (ref Frühbeck G, Gómez-Ambrosi J, Salvador J. Leptin-induced lipolysis opposes the tonic inhibition of endogenous adenosine in white adipocytes. FASEB J. 2001 Feb;15(2):333-40  // Frühbeck G, Gómez-Ambrosi J. Modulation of the leptin-induced white adipose tissue lipolysis by nitric oxide. Cell Signal. 2001 Nov;13(11):827-33). Moreover, in relation to the role of glucose tolerance and dysfunctional adiposity, it would be worthwhile mentioning the relevance of the leptin-adiponectin ratio (ref  Frühbeck G, Catalán V, Rodríguez A, Ramírez B, Becerril S, Salvador J, Portincasa P, Colina I, Gómez-Ambrosi J. Involvement of the leptin-adiponectin axis in inflammation and oxidative stress in the metabolic syndrome. Sci Rep. 2017 Jul 26;7(1):6619).

Response 8: We are sorry for the omission of this part of the content. Research on the effect of leptin on lipolysis has made great progress in the past two decades. In response to the references you provided, in the discussion section, we have further added other effects of leptin on lipolysis in addition to the mechanism of action in this study, so as to make the discussion of this manuscript more complete. The revised paragraphs include lines 374-382 on page 16. Thank you again for your careful review.

Point 9: Discussion: furthermore, as regards body composition and effect on body temperature it is also important to mention the effect of leptin on myogenesis (ref Rodríguez A, Becerril S, Méndez-Giménez L, et al. Leptin administration activates irisin-induced myogenesis via nitric oxide-dependent mechanisms, but reduces its effect on subcutaneous fat browning in mice. Int J Obes (Lond). 2015 Mar;39(3):397-407).

Response 9: Thank you very much for your valuable suggestions and references. The effect of leptin on myogenesis and this reference has been supplemented in the introduction. Please refer to the revised manuscript for details.

Point 10: Mat & Methods: the methodology is adequately described.

Response 10: We feel great thanks for your professional suggestion on our article. We also checked the materials and methods again to ensure that they were adequately described.

References

  1. Bakshi, A.; Singh, R.; Rai, U., Trajectory of leptin and leptin receptor in vertebrates: Structure, function and their regulation. Comparative Biochemistry and Physiology Part B: Biochemistry and Molecular Biology 2022, 257, 110652.
  2. Tung, Y. C. L.; Yeo, Giles S. H.; O’Rahilly, S.; Coll, Anthony P., Obesity and FTO: Changing Focus at a Complex Locus. Cell metabolism 2014, 20, (5), 710-718.

Reviewer 3 Report

The aim of this study was to examine the mechanism through which leptin stimulates lipolysis in subcutaneous pig adipose tissue. Leptin was administered to piglets at a dose of 1 mg/kg/day for 4 weeks or was added to adipocyte culture in vitro. By using combined pharmacological and genetic approaches, authors have demonstrated that leptin increases the expression of FTO, reduces total RNA and perilipin-5 mRNA m6 adenosine methylation, increases the expression of lipolysis-associated genes and mitochondrial complex proteins. Experiments with FTO overexpression/silencing demonstrated the key role of this protein in regulating RNA methylation. Furthermore, reducing effect of leptin on Pln-5 mRNA methylation results in the up-regulation of Pln5 protein expression, presumably by increasing translation efficacy without affecting mRNA level..

The topic and the results are of interest, however, there are also some concerns to be addressed.

  • Abstract: the methods should be briefly described.
  • Introduction should be re-written to become more informative for those being not the specialists in the leptin field. The 2-3 sentence introduction about leptin itself would be valuable. The role of FTO and perilipins should also be briefly mentioned.
  • Line 28: the sentence that “leptin reduces the appetite of obese individuals” is questionable. Obesity is often associated with leptin resistance and obese animals/humans remain hyperphagic despite higher leptin level. Similarly (lines 33-34), leptin is not “an effective hormone for alleviating obesity and its related complications”. Treatment with exogenous leptin is effective only in rare cases of obesity associated with leptin deficiency. In addition, many studies suggest that hyperleptinemia associated with obesity contributes to the development of metabolic and other complications of obesity.
  • Line 72 and 76: the methods of measuring intramuscular fat content and body temperature should be described in the Method section.
  • According to figure 1A and B, leptin treatment increased the expression of leptin and its receptor. In contrast, in the text (lines 69-70) authors state that leptin treatment reduced their expression. This discrepancy should be clarified.
  • The legend for figure 1 should be modified. The title of the figure is not appropriate because the effect of leptin on lipolysis itself was not examined. Only the expression of some proteins involved in lipolysis was measured. The phrase: “Effect of leptin treatment on…” should not be repeated for every panel separately. The legend should modified to, for example: “Effect of leptin treatment on: (A)…, (B)…, etc.”
  • Line 93, the phrase “Pln5 downstream lipolysis genes” – do the authors mean that Pln5 regulates the expression of these genes? What underlying mechanism is suggested?
  • It should be clearly stated in figure legends which results were obtained in vitro and which in vivo. For example, line 114 (legend for figure 2), “pig adipocytes” – it is unclear if adipocytes were treated with leptin in vitro or were isolated from pigs treated with leptin in vivo.
  • Line 122: the conclusion that FTO was involved in the effect of leptin on Pln5 is preliminary in this place because the results of FTO overexpression/silencing are presented in the subsequent section.
  • Section 4.4: how specific are cycloleucine and betaine in affecting RNA methylation? What about their effects on the methylation of DNA or other targets?
  • Section 4.15: more details about qRT-PCR should be provided. In particular, time and temperatures of consecutive cycle phases should be presented.
  • Section 4.15: the method of densitometry/quantification of Western blot results should be described.
  • Statistical analysis: was normality of data distribution verified to warrant using parametric tests? In addition, why ANOVA (designed for comparison of 3 or more groups) was used for the experiments in which only two groups (control and leptin-treated) were included?
  • Most results are very similar in the in vivo and in vitro experiments. On the other hand, it is well known that leptin stimulates sympathetic nervous system through the central mechanism when administered in vivo and sympathetic system potently regulates adipose tissue lipolysis. How the involvement of sympathetic system could affect the results obtained in the in vivo part of this study?

Author Response

Dear reviewer,

We would like to thank you very much for your valuable comments and good suggestions that greatly helped to improve our manuscript. Thank you very much for your time and efforts. We have studied comments carefully and have made correction which we hope meet with approval. At the same time, we also invited a native English-speaking colleague to revise the grammar of the article and make the appropriate corrections. Revised portion are highlighted in the track changes mode in MS Word. We hope that our revision can meet your requirements. The main corrections in the paper and the responds to the reviewer’s comments are as flowing:

Point 1: Abstract: the methods should be briefly described.

Response 1: Thank you for your reminding. We have added the content of the piglet-related trial groups/design in the abstract. Please refer to the revised manuscript for details.

Point 2: Introduction should be re-written to become more informative for those being not the specialists in the leptin field. The 2-3 sentence introduction about leptin itself would be valuable. The role of FTO and perilipins should also be briefly mentioned.

Response 2: Thank you for your careful review. We are deeply sorry for the insufficient and comprehensive description of leptin, FTO and perilipins in the introduction. In response to your comments, we have further revised the relevant content in the introduction. The revised paragraphs include lines 34-36 on page 2, lines 37-47 on page 3.

Point 3: Line 28: the sentence that “leptin reduces the appetite of obese individuals” is questionable. Obesity is often associated with leptin resistance and obese animals/humans remain hyperphagic despite higher leptin level. Similarly (lines 33-34), leptin is not “an effective hormone for alleviating obesity and its related complications”. Treatment with exogenous leptin is effective only in rare cases of obesity associated with leptin deficiency. In addition, many studies suggest that hyperleptinemia associated with obesity contributes to the development of metabolic and other complications of obesity.

Response 3:  Thank you for your reminder. We are very sorry for the inaccurate and objective description of leptin in the introduction. We deleted these sentences and made a lot of revisions based on the latest reports. The revised paragraphs include lines 37-39,40-47 and 50-51 on page 3.

Point 4: Line 72 and 76: the methods of measuring intramuscular fat content and body temperature should be described in the Method section.

Response 4:  Thank you for your careful review. We have added corresponding content about intramuscular fat and body temperature measurement in 2.1 Animals and Samples of Materials and Methods. Please refer to the revised manuscript for details.

Point 5: According to figure 1A and B, leptin treatment increased the expression of leptin and its receptor. In contrast, in the text (lines 69-70) authors state that leptin treatment reduced their expression. This discrepancy should be clarified.

Response 5:  Thank you for your valuable suggestions to improve the quality of our manuscript. According to the data, leptin treatment significantly increased the level of corresponding indicators, but we made a wrong description in the text. We have changed "reduced" to "increased" in the text.

Point 6: The legend for figure 1 should be modified. The title of the figure is not appropriate because the effect of leptin on lipolysis itself was not examined. Only the expression of some proteins involved in lipolysis was measured. The phrase: “Effect of leptin treatment on…” should not be repeated for every panel separately. The legend should modified to, for example: “Effect of leptin treatment on: (A)…, (B)…, etc.”

Response 6:  We feel great thanks for your professional suggestion on our article. In response to your suggestion, we have modified the title of Figure 1 to: “Leptin improves the body fat composition and weight of piglets by up-regulating the expression of Plin5”. In addition, the phrase “Effect of leptin treatment on...” in all figures legends has also been appropriately modified based on the example you provided. Please refer to the revised manuscript for details.

Point 7: Line 93, the phrase “Pln5 downstream lipolysis genes” – do the authors mean that Pln5 regulates the expression of these genes? What underlying mechanism is suggested?

Response 7:  Thank you for your careful review. PPARγ, ATGL, HSL and LPL are all genes involved in regulating lipolysis. It has been reported that Plin5 participates in lipolysis through interaction with ATGL or HSL [1-3]. At the same time, PPARγ also induces the accumulation of Plin5 to improve lipid metabolism disorders [4]. Our description of "Pln5 downstream lipolysis genes" in the text is inaccurate. The underlying mechanism we want to describe is that these genes interact to regulate lipid metabolism. Therefore, we deleted the phrase “Pln5 downstream”.

Point 8: It should be clearly stated in figure legends which results were obtained in vitro and which in vivo. For example, line 114 (legend for figure 2), “pig adipocytes” – it is unclear if adipocytes were treated with leptin in vitro or were isolated from pigs treated with leptin in vivo.

Response 8: I'm sorry we didn't make a clear description about treatment of leptin in vitro and vivo. Thank you very much for reminding. We make supplementary explanations for the relevant content in figure legends. Please refer to the revised manuscript for details.

Point 9: Line 122: the conclusion that FTO was involved in the effect of leptin on Pln5 is preliminary in this place because the results of FTO overexpression/silencing are presented in the subsequent section.

Response 9: Thank you for your reminder. We have modified the description of this line and pointed out that this is a preliminary conclusion. Please refer to the revised manuscript for details.

Point 10: Section 4.4: how specific are cycloleucine and betaine in affecting RNA methylation? What about their effects on the methylation of DNA or other targets?

Response 10: Thank you for your careful review. The methylation inhibitors and methylation donors used in this study are based on others' methods [5, 6], and they are also specific methylation inhibitors and donors. A large number of studies have reported that cycloleucine is a specific and reversible inhibitor of RNA methylation [6-9]. During the cycloleucine treatment, the rate of biosynthesis of hnRNA and its subsequent polyadenylation were only slightly reduced as compared with untreated cells. A significant lag-time in the cytoplasmic appearance of poly(A)+ undermethylated molecules was observed, in parallel with a transient shift in the average size of hnRNA towards higher molecular weight.  Cycloleucine has an effect on DNA methylation enzymes, but it is not the most effective compound for inhibiting DNA methylation. There is no difference between delayed methylation reaction and non-delayed methylation reaction [10]. Therefore it is widely used as an m6A methylation inhibitor. Betaine is not only a metabolite of choline but also a methyl group donor that participates in methylation [11]. Methyl donors affect the level of methylation, including DNA and RNA methylation. The methyl transfer reaction of betaine is a single carbon metabolism via the methionine cycle. In this reaction, BHMT catalyzes the addition of a methyl group from betaine to homocysteine to form methionine, which is subsequently converted to dimethylglycine (DMG). DMG has two available methyl groups and is possibly degraded to sarcosine and ultimately to glycine, and produce S-adenosylmethionine (SAM). SAM is the main methylating agent. After demethylation, SAM can be converted to S-adenosyl homocysteine (SAH). The ratio of SAM:SAH affects various SAM-dependent methyltransferases [12]. Therefore, betaine is commonly used as a methylation-specific donor and plays an important role in methylation metabolism.

Point 11: Section 4.15: more details about qRT-PCR should be provided. In particular, time and temperatures of consecutive cycle phases should be presented.

Response 11: Thank you very much for your valuable suggestions. We have added the detailed information of RT-PCR in the corresponding text of the Material and Methods. The reaction program is set to 95℃, 10s (pre-change period), 95℃, 5s, 60℃, 30s (PCR reaction period), a total of 40 cycles.

Point 12: Section 4.15: the method of densitometry/quantification of Western blot results should be described.

Response 12: Thank you very much for your valuable suggestions in our manuscript. We have further described the densitometry/quantitative method of Western blot results in Materials and Methods. Please refer to the revised manuscript for details.

Point 13: Statistical analysis: was normality of data distribution verified to warrant using parametric tests? In addition, why ANOVA (designed for comparison of 3 or more groups) was used for the experiments in which only two groups (control and leptin-treated) were included?

Response 13: We feel great thanks for your professional suggestion on our manuscript. We are sorry that the description of statistical analysis is not specific and complete. In this regard, we have modified the corresponding part of the text. Please refer to the revised manuscript for details. Kolmogorov–Smirnov tests were used to test the data for normality in the SAS9.4 software (SAS Inst. Inc., Cary, NC, USA). The control and leptin-treated data were analyzed by independent samples T test for significance. The other in vitro cell treatment experiments of 3 groups or more were analyzed by one-way ANOVA.

Point 14: Most results are very similar in the in vivo and in vitro experiments. On the other hand, it is well known that leptin stimulates sympathetic nervous system through the central mechanism when administered in vivo and sympathetic system potently regulates adipose tissue lipolysis. How the involvement of sympathetic system could affect the results obtained in the in vivo part of this study?

Response 14: We very much agree with your ideas. As you said, Leptin is a negative feedback signal, which is mainly produced by adipose tissue, and the circulating concentration is related to the size of body fat storage. Since adipocytes express leptin receptors, leptin can directly affect the metabolism of adipocytes. At the same time, leptin is also innervated by sympathetic nerves, so leptin can also indirectly change the metabolism of adipocytes. Published studies have shown that leptin directly activates the leptin receptor of adipocytes to regulate intracellular lipid metabolism [13-15]. Consistently, injection of leptin in the body can inhibit fat production, increase triglyceride hydrolysis, and increase fatty acid and glucose oxidation [16-18]. This is the same effect on adipocytes. This is consistent with our results in vivo and in vitro. The activation of the central leptin receptor contributes to the development of a catabolic state in adipocytes. In the participation of the sympathetic nervous system, it reduces the size of white fat depots by inhibiting cell proliferation both through induction of inhibitory circulating factors and by contributing to sympathetic tone which suppresses adipocyte proliferation. Therefore, the results of this study in vivo must have the effect of sympathetic nerves. However, the focus of this study is not to consider sympathetic innervation to improve metabolism, but to focus on the intracellular effects of leptin directly binding to the adipocyte leptin receptor. Thank you very much for your valuable comments.

References

  1. Gallardo-Montejano, V. I.; Yang, C.; Hahner, L.; McAfee, J. L.; Johnson, J. A.; Holland, W. L.; Fernandez-Valdivia, R.; Bickel, P. E., Perilipin 5 links mitochondrial uncoupled respiration in brown fat to healthy white fat remodeling and systemic glucose tolerance. Nature communications 2021, 12, (1), 3320-3320.
  2. Whytock, K. L.; Shepherd, S. O.; Wagenmakers, A. J. M.; Strauss, J. A., Hormone-sensitive lipase preferentially redistributes to lipid droplets associated with perilipin-5 in human skeletal muscle during moderate-intensity exercise. J Physiol 2018, 596, (11), 2077-2090.
  3. Granneman, J. G.; Moore, H.-P. H.; Mottillo, E. P.; Zhu, Z.; Zhou, L., Interactions of perilipin-5 (Plin5) with adipose triglyceride lipase. The Journal of biological chemistry 2011, 286, (7), 5126-5135.
  4. Tian, S.; Lei, P.; Teng, C.; Sun, Y.; Song, X.; Li, B.; Shan, Y., Targeting PLIN2/PLIN5-PPARγ: Sulforaphane Disturbs the Maturation of Lipid Droplets. Molecular nutrition & food research 2019, 63, (20), e1900183.
  5. Wang, X.; Zhu, L.; Chen, J.; Wang, Y., mRNA m⁶A methylation downregulates adipogenesis in porcine adipocytes. Biochemical and biophysical research communications 2015, 459, (2), 201-207.
  6. Kang, H.; Zhang, Z.; Yu, L.; Li, Y.; Liang, M.; Zhou, L., FTO reduces mitochondria and promotes hepatic fat accumulation through RNA demethylation. Journal of cellular biochemistry 2018, 119, (7), 5676-5685.
  7. Bachellerie, J. P.; Amalric, F.; Caboche, M., Biosynthesis and utilization of extensively undermethylated poly(A)+ RNA in CHO cells during a cycloleucine treatment. Nucleic acids research 1978, 5, (8), 2927-43.
  8. Dimock, K.; Stolzfus, C. M., Cycloleucine blocks 5'-terminal and internal methylations of avian sarcoma virus genome RNA. Biochemistry 1978, 17, (17), 3627-32.
  9. Caboche, M.; Bachellerie, J. P., RNA methylation and control of eukaryotic RNA biosynthesis. Effects of cycloleucine, a specific inhibitor of methylation, on ribosomal RNA maturation. European journal of biochemistry 1977, 74, (1), 19-29.
  10. Woodcock, D. M.; Adams, J. K.; Allan, R. G.; Cooper, I. A., Effect of several inhibitors of enzymatic DNA methylation on the in vivo methylation of different classes of DNA sequences in a cultured human cell line. Nucleic acids research 1983, 11, (2), 489-99.
  11. Zhao, G.; He, F.; Wu, C.; Li, P.; Li, N.; Deng, J.; Zhu, G.; Ren, W.; Peng, Y., Betaine in Inflammation: Mechanistic Aspects and Applications. Front Immunol 2018, 9, 1070-1070.
  12. Lever, M.; Slow, S., The clinical significance of betaine, an osmolyte with a key role in methyl group metabolism. Clinical Biochemistry 2010, 43, (9), 732-744.
  13. Harris, R. B. S., Direct and indirect effects of leptin on adipocyte metabolism. Biochim Biophys Acta 2014, 1842, (3), 414-423.
  14. Wang, J.; Ge, J.; Cao, H.; Zhang, X.; Guo, Y.; Li, X.; Xia, B.; Yang, G.; Shi, X., Leptin Promotes White Adipocyte Browning by Inhibiting the Hh Signaling Pathway. Cells 2019, 8, (4).
  15. Liu, Z.; Gan, L.; Zhou, Z.; Jin, W.; Sun, C., SOCS3 promotes inflammation and apoptosis via inhibiting JAK2/STAT3 signaling pathway in 3T3-L1 adipocyte. Immunobiology 2015, 220, (8), 947-53.
  16. Dallner, O. S.; Marinis, J. M.; Lu, Y. H.; Birsoy, K.; Werner, E.; Fayzikhodjaeva, G.; Dill, B. D.; Molina, H.; Moscati, A.; Kutalik, Z.; Marques-Vidal, P.; Kilpeläinen, T. O.; Grarup, N.; Linneberg, A.; Zhang, Y.; Vaughan, R.; Loos, R. J. F.; Lazar, M. A.; Friedman, J. M., Dysregulation of a long noncoding RNA reduces leptin leading to a leptin-responsive form of obesity. Nature medicine 2019, 25, (3), 507-516.
  17. Zhao, S.; Zhu, Y.; Schultz, R. D.; Li, N.; He, Z.; Zhang, Z.; Caron, A.; Zhu, Q.; Sun, K.; Xiong, W.; Deng, H.; Sun, J.; Deng, Y.; Kim, M.; Lee, C. E.; Gordillo, R.; Liu, T.; Odle, A. K.; Childs, G. V.; Zhang, N.; Kusminski, C. M.; Elmquist, J. K.; Williams, K. W.; An, Z.; Scherer, P. E., Partial Leptin Reduction as an Insulin Sensitization and Weight Loss Strategy. Cell metabolism 2019, 30, (4), 706-719.e6.
  18. Zhou, Y.; Yu, X.; Chen, H.; Sjöberg, S.; Roux, J.; Zhang, L.; Ivoulsou, A. H.; Bensaid, F.; Liu, C. L.; Liu, J.; Tordjman, J.; Clement, K.; Lee, C. H.; Hotamisligil, G. S.; Libby, P.; Shi, G. P., Leptin Deficiency Shifts Mast Cells toward Anti-Inflammatory Actions and Protects Mice from Obesity and Diabetes by Polarizing M2 Macrophages. Cell metabolism 2015, 22, (6), 1045-58.

Reviewer 4 Report

Overall the manuscript was well written although there are few major problems needed to be addressed. 

1) In figure-1 the leptin  and its receptors are shown as they are increased after treatment with leptin but in text it was written as it is reduced.

2) Section 2.4 in results the text referring to figure 6 but there is none to be found in the text.

3) In section 2.4 the NAD+/ NADH levels were mentioned to be altered but the figure was not there and also no methods to show how the experiment was performed.

4) Please provide table for primers used in the RT-PCR.

Author Response

We would like to thank you very much for your valuable comments and good suggestions that greatly helped to improve our manuscript. Thank you very much for your time and efforts. We have studied comments carefully and have made correction which we hope meet with approval. At the same time, we also invited a native English-speaking colleague to revise the grammar of the article and make the appropriate corrections. Revised portion are highlighted in the track changes mode in MS Word. We hope that our revision can meet your requirements. The main corrections in the paper and the responds to the reviewer’s comments are as flowing:

Point 1: In figure-1 the leptin and its receptors are shown as they are increased after treatment with leptin but in text it was written as it is reduced.

Response 1: Thank you for your valuable suggestions to improve the quality of our manuscript. According to the data, leptin treatment significantly increased the level of corresponding indicators, but we made a wrong description in the text. We have changed "reduced" to "increased" in the text.

Point 2: Section 2.4 in results the text referring to figure 6 but there is none to be found in the text.

Response 2: Thank you for your careful review. There are a total of 7 figures in our manuscript, but two figures in the PDF version of the manuscript are not displayed properly. We will contact the editor to deal with the situation. We are very sorry for the trouble to you.

Point 3: In section 2.4 the NAD+/ NADH levels were mentioned to be altered but the figure was not there and also no methods to show how the experiment was performed.

Response 3: We are very sorry for the trouble to you. Our manuscript has a total of 7 figures. The NAD+/NADH data is shown in Figure 6. However, the last two figures of the PDF version of the manuscript failed to display properly. We will contact the editor to deal with the situation. Simultaneously, we have also supplemented the NAD+/NADH detection method in the materials and methods. Please refer to the revised manuscript for details.

Point 4: Please provide table for primers used in the RT-PCR.

Response 4: Thank you for your careful review. All primers for RT-PCR are placed in Supplementary Table S1.

Round 2

Reviewer 4 Report

It is important to add the western blot for figure 6 to support the data as pcDNA3.1 and pc-Plin5 caused expression difference is not very significant.

Author Response

We are very grateful for your careful review of our manuscript and your valuable suggestions. We have made further data supplements and revisions based on your suggestions in the corresponding parts of the manuscript. We hope you are satisfied with the revised manuscript and agree to accept it. The main corrections in the paper and the responds to the reviewer’s comments are as flowing:

Point 1: It is important to add the western blot for figure 6 to support the data as pcDNA3.1 and pc-Plin5 caused expression difference is not very significant.

Response 1: We are very sorry for not including western blot analysis in Figure 6. Based on your comments, we analyzed the protein levels of genes whose expression changes caused by pcDNA3.1 and pc-Plin5. Here, we used the protein samples stored earlier to detect the expression levels of PPARGC1A, NDUFB8, Plin5, DGAT2 and CPT1a, and further added the results to Figures 6C and 6M. For details, please refer to our revised manuscript.